# Children’s SARS-CoV-2 Infection and Their Vaccination

**DOI:** 10.3390/vaccines11020418

**Published:** 2023-02-12

**Authors:** Sneh Lata Gupta, Rohit Tyagi, Atika Dhar, Neelam Oswal, Ankita Khandelwal, Rishi Kumar Jaiswal

**Affiliations:** 1National Institute of Immunology, New Delhi 110067, India; 2College of Veterinary Medicine, Huazhong Agricultural University, Wuhan 430070, China; 3Jhalawar Medical College, Jhalawar 326001, India; 4Department of Cancer Biology, Cardinal Bernardin Cancer Center, Stritch School of Medicine, Loyola University Chicago, Maywood, IL 60153, USA

**Keywords:** SARS-CoV-2, MIS-C, Omicron, EUA

## Abstract

SARS-CoV-2, a novel coronavirus, causes respiratory tract infections and other complications in affected individuals, and has resulted in numerous deaths worldwide. The unprecedented pace of its transmission worldwide, and the resultant heavy burden on healthcare systems everywhere, prompted efforts to have effective therapeutic strategies and vaccination candidates available to the global population. While aged and immunocompromised individuals form a high-risk group for COVID-19 and have severe disease outcome, the rate of infections among children has also increased with the emergence of the Omicron variant. In addition, recent reports of threatening SARS-CoV-2-associated complications in children have brought to the forefront an urgent necessity for vaccination. In this article, we discuss the current scenario of SARS-CoV-2 infections in children with a special focus on the differences in their immune system response as compared to adults. Further, we describe the various available COVID-19 vaccines, including the recent bivalent vaccines for children, in detail, intending to increase willingness for their acceptance.

## 1. Introduction

SARS-CoV-2 has displayed a very high rate of spread since December 2019, resulting in an enormous global death toll [1,2,3,4]. A total of 754 million confirmed cases and 6.81 million deaths due to SARS-CoV-2 infections have been reported until now, as per the World Health Organization (WHO) epidemiological report (https://covid19.who.int/, accessed on 30 November 2022). The emergence of new variants of SARS-CoV-2 such as Alpha, Beta, Delta, and the most recent Omicron variant and subvariants have posed a challenge for the healthcare system due to the high transmissibility [5,6,7], contributing to the need to design effective treatment measures. According to published data, elderly individuals and those with co-morbidities have shown a higher incidence of fatality, and therefore are at a higher risk due to the virus infection [8,9]. On the other hand, SARS-CoV-2 infections in children have mostly been reported to be mild, with a low fatality rate [10,11,12,13].

A major protective measure taken to reduce COVID-19 infections in children has been the closure of schools and play areas, but these measures have drawbacks and cannot be implemented long-term [14]. Despite the early notion of a low number of cases in children, an increase in the number of infections and hospitalizations due to Delta and the subsequent Omicron variants of SARS-CoV-2 has been observed [15]. In addition, children can form a point of contact with COVID-19 infection for their family members, which might constitute individuals in the high-risk group. The vaccination drive against COVID-19 in adults has helped in controlling disease transmission and severity to a large extent [16]. Therefore, it is also necessary to extend this protection to children by promoting their vaccination. Vaccination against many life-threatening infections is commonly administered to children during the first few years after birth, and including COVID-19 vaccination might be a significant step.

A majority of the available literature deals with SARS-CoV-2 infection in adults and their vaccination, though limited research has been carried out in cases of children. This is largely explained by preliminary results which showed that children are either asymptomatic or have mild symptoms after SARS-CoV-2 infection. However, it does not rule out the importance of vaccination among children to achieve herd immunity and risk of getting exposed with the continuous emergence of newer SARS-CoV-2 variants. Vaccination of children requires parental consent, and vaccine hesitancy among parents exists due to limited literature availability, fewer numbers of or lack of available vaccines, lack of FDA approved bivalent vaccine for kids aged 5 years or less, and the available vaccines being under the Emergency Use Authorization (EUA) category and not fully licensed for vaccination in children in the public domain as compared to adults. Infants younger than 6 months are dependent on maternal antibody transfer, as no vaccine is available for them. Therefore, an infant will not receive protective antibodies unless the mother is vaccinated or possesses antibodies against SARS-CoV-2 after an infection. Additionally, cases of Multisystem Inflammatory Syndrome in children (MIS-C) have also been reported, though the exact causative mechanism has not yet been understood, which has hindered the development of a universal treatment for MIS-C. Children also differ from adults in terms of innate, adaptive, and mucosal immune responses. Hence, the immune response to an infection or vaccination in adults is different from that observed in children. All these challenges make children a special cohort and highlight the need for an increase in the availability of literature in the public domain. This review encourages parents to get their children vaccinated, and the researchers and policy makers to assess pediatric vaccination safety, immunogenicity, and efficacy data and facilitate its proper dispersal. This review also reflects the importance of finding new alternatives of vaccine availability, their dose, route of administration, etc., that might differ from adults in the future.

COVID-19 is an infectious disease that is transmitted mainly via respiratory droplets. The SARS-CoV-2 virus has also been reported to be sustained on surfaces, and hence maintaining social distancing, use of masks, and hand sanitizers were recommended to prevent community transmission [17,18,19,20,21]. In a family setting, there were reported cases of family transmission of SARS-CoV-2 infection and related disease symptoms. A case study conducted in China, including 14 families and nine children, reported that adults in these families were SARS-CoV-2 PCR positive and had moderate to severe symptoms, and all nine children were also positive but had mild symptoms (3) or were asymptomatic (6) compared to adults. Similarly, other reports also showed that children can be potential carriers of this disease and become infected via family transmission [22,23,24]. In children, severe disease did not positively correlate with viral load [25]. This suggests that even asymptomatic children, or those with mild symptoms, could have higher viral titers and serve as a potential carrier for community disease transmission and the emergence of SARS-CoV-2 viral variants.

Newborn infants can develop SARS-CoV-2 infection, and around 2–3% of maternal to fetal vertical transfer (intrauterine transfer) of infection (lower risk of incidence) has been reported [26,27,28,29]. Infants receive maternal antibodies (vertical transmission) from their mother from the placenta and during the lactation period via breast milk [30,31,32]. A recent study suggested that antibodies are transferred from vaccinated or SARS-CoV-2-infected mothers to their babies via placental transfer. The transferred antibody titers sharply decrease from birth until 6 months of age [33,34,35,36,37], as shown in Figure 1. Pregnant women vaccinated with the mRNA vaccine show better antibody-mediated function and Fc receptor binding than the adenovirus-based vaccine. Among the three trimesters during pregnancy, vaccination in the first or third trimester gives a better immune response than vaccination in the second trimester. The transfer efficiency of antibodies from mother to fetus is also higher in the first and second trimesters compared to the third trimester when tested in both maternal and umbilical cord blood sites [38]. Maternal antibodies are also transferred via breastfeeding during lactation [39]. Children have been shown to have a durable immune response after COVID-19 infection. As reported, children showed long-term Receptor Binding Domain (RBD) binding antibody responses to SARS-CoV-2 infection in various age groups up to 10 months post-COVID-19 mild infection [40]. It was observed that very young kids, such as children <5 years of age, who were affected by severe acute COVID-19, and later hospitalized, had a greater reduction in neutralizing antibodies to SARS-CoV-2 variants as compared to children >5 years of age. This report also confirmed that convalescent COVID-19 and MIS-C affected pediatric cohorts showed higher neutralization titers as compared to acutely infected COVID-19 populations [41]. Studies have shown that kids younger than 5 years of age had a higher incidence rate during the emergence of the Omicron variant in comparison to the Delta variant. This suggested that younger kids can transmit the Omicron variant infection at a higher rate, but have less severe clinical outcomes. Among these kids, those belonging to the age group of 0–2 years had a higher monthly incidence rate of COVID-19 as compared to those of 3–4 years of age during the emergence of the Omicron variant [42,43,44].

Various studies have explained that the reasons children have a milder infection are strong innate responses due to higher levels of IFN-gamma and IL-17A, and lower levels of TNF-alpha and IL-6 in serum. SARS-CoV-2 infected children also have a lower adaptive immune response as shown by lower memory T cells, Fc gamma receptor levels, lesser ADCP (antibody-dependent cellular phagocytosis) reactions, and reductions in neutralizing antibody responses than adults [45]. The innate immune response in the nasal mucosa of children is also stronger and more vigorous than adults. Nasopharyngeal swabs from COVID-19-infected children and adults were compared; it was found that children displayed higher gene expression related to IFN and NLRP3 inflammasome signaling [46,47]. Children also showed higher basal level gene expression of pattern recognition receptors (PRRs), such as MDA5 and RIG-1, in innate immune cells in upper respiratory airways than adults, which explains the robust anti-viral innate immune response in children compared to adults [48]. The adaptive immune system of children is naïve compared to that of adults. As a result, they have a higher frequency of naïve T cells, especially cytotoxic T cells and NK cells, and less clonal expansion in the T cell repertoire. Together, these factors possibly contribute to the absence of an exacerbated immune reaction and hyperinflammation post-SARS-CoV-2 infection in children [49]. Children tend to have higher clonotype diversity and enriched naïve B cell and T cell populations as compared to adults. On the other hand, adults have a higher frequency of systemic IFN-induced immune cells such as B cells, T cells, NK cells, and monocytes which induce IFN in blood, as well as enrichment of cytotoxic immune cells [50]. One in vitro assay assessing SARS-CoV-2 specific IFN-gamma producing T cells showed that they are higher in adults compared to children, in cases of mild to moderate SARS-CoV-2 infection [51]. Children also developed a stronger spike specific B cell mediated antibody response, as well as T cell response, which was also persistent and durable (more than 6 months), as compared to adults (faster decay), post SARS-CoV-2 infection [52,53]. Both acute and memory CD4+ T cells and CD8+ T cell response is lower in children than adults [54].

There are also other factors contributing to lower susceptibility to SARS-CoV-2 infection in children. The first factor is the lower intensity of exposure to SARS-CoV-2, since the family dynamics established during the pandemic were intended to protect the children. The second factor is cross reactivity, since children tend to have a higher frequency of recurrent and concurrent viral exposure as well as vaccinations. These repeated and multiple viral infections and various early-stage vaccinations lead to an ongoing state of activation of the innate immune system [55,56]. Similarly, children have a higher exposure to helminth infections than adults, which causes modulation of host inflammatory components [57]. Besides these factors, several other immunological factors are different in cases of adults than in children. The first is that adults have an alteration in their endothelial function and coagulation [58]. The second is the difference in enzymatic density and affinity on respiratory mucosal epithelial cells. These enzymes are angiotensin converting enzyme (ACE2) and transmembrane protease, serine 2 (TMPRSS2) [59]. In addition, adults have a higher immune senescence and chronic inflammation rate than children, and a higher prevalence of comorbidities and underlying chronic conditions [60]. In summary, all these differences in innate and adaptive immune responses reflect why children show a distinct immune response post-SARS-CoV-2 infection than adults.

## 2. Vaccine Hesitancy in Parents to Have Their Children Vaccinated

A major challenge in containing the spread of the SARS-CoV-2 virus is to have most of the population vaccinated, potentially leading to lower disease severity and transmission [61]. While various countries have taken measures to promote vaccination, such as the provision of free vaccines, the percentage of people taking the vaccines in developed countries is not 100%. As reported in latest data on 11 February 2023 69.4% world population has received one dose of vaccine but only 26.4% in low income countries got their one dose. (https://ourworldindata.org/covid-vaccinations, accessed on 30 November 2022) [62]. This indicates a certain degree of hesitancy among the adult population regarding the risks of taking the vaccine, and this is largely expected to affect the vaccination rate in children. Several studies have attempted to understand the factors that affect parents’ decision to get their children vaccinated based on surveys. One study conducted among parents of adolescent children of 16+ years old showed that the most common reasons for hesitancy among parents for vaccinating their children were concerns about long-term side effects and possible negative effects [63]. Another study tried to review the reasons affecting vaccine acceptance among low-income group parents with children in the age group of newborns to 17 years old. In the study, some parents described their unwillingness to vaccinate their children due to milder forms of the disease occurring in the children. Other factors also noted as reasons for hesitancy in the study include possible negative side effects of the vaccine, lack of knowledge about the long-term effects, and the speed of the vaccine development [64]. Furthermore, a survey-based study also found that the vaccine hesitancy was also influenced by the gender of the parents, being higher in females, their economic status being higher among low-income groups, and their political beliefs. This study also noted the primary concern of parents was the safety of the vaccine in comparison to its effectiveness [65]. Another factor influencing the decision of vaccination of children is the fear of vaccines originating from abroad. In addition, this study also noted that the anxiety of parents about negative vaccine side-effects and lack of knowledge about vaccine effectiveness were the most common reasons for hesitancy in taking the vaccine [66].

Many parents rely on online sources of information, which very often are biased and foster disbelief in the parents about the risks of vaccinating their children [67,68]. To overcome these challenges, better spread of correct information through online channels, and promotion of vaccine uptake by medical practitioners and government agencies are important measures. Also, vaccination against other diseases in children has rarely shown long-term negative effects, suggesting that COVID-19 vaccines have a very low risk factor when considered long term [69].

## 3. Multi-Inflammatory Syndrome in Children

In children, COVID-19 infections are mild or asymptomatic and mostly self-limiting at the time of acute infection. Post 3–4 weeks from COVID-19 infection, some children develop multiorgan hyperinflammatory syndrome, known as MIS-C. It is a rare disease condition that afflicts both children (MIS-C) and adults (MIS-A). In April 2020, a few cases of MIS-C were reported in Europe. It is a systemic disorder affecting multiple organs. Fever, gastrointestinal, abdominal pain are key diagnostic symptoms of this disorder. In addition to these symptoms, cardiac disease is the most common comorbidity associated with MIS-C, along with respiratory and neurological disorders. It should also be understood that while disease severity in children has been reported to be mild, a serious form of the disease, i.e., MIS-C, has also been reported [70,71]. The most frequent symptoms observed in this syndrome are depicted in Figure 2. Due to the similarity of MIS patient symptoms with septic shock, toxic shock syndrome (TSS) and Kawasaki disease (KD), the initial treatment prescribed to treat MIS-C was similar to TSS and KD, but later MIS-C was found to be associated with COVID-19 disease [72,73]. The majority of MIS-C cases require hospitalization and intensive care unit (ICU) admission. Despite severe symptoms, mortality rate was as low as 1.9%. It has been reported that MIS-C occurrence dominates in males and black ethnic groups [74]. MIS-C patients suffer from a hyperinflammatory condition, which might occur due to an increase in the amount of activated immune cells such as neutrophils, monocytes, DCs, NK cells, B and T cells, and flares of cytokines and chemokines that cause vascular patrolling of these active cells to organs, and in some cases a higher amount of autoantibodies cumulatively affected multiple organs [75,76,77]. One interesting observation was that Omicron infected children developed lesser MIS-C, especially post vaccination, or in cases of reinfection as compared to the Delta variant, although the MIS-C phenotype was mostly similar. This may be due to the modulation of immune system by vaccination or previous infection to cause lesser hyperinflammation. Thus, children in the Omicron wave had a lesser risk of MIS-C development than in the earlier Delta or Alpha variant waves [78,79]. The exact mechanism and early disease biomarkers for MISC-C are still undetermined, and the paradox of complex hyperinflammation remains largely unsolved.

## 4. Status of Vaccination in Children

Safe, immunogenic, and effective vaccines for children of all ages are currently needed, and only a few are approved for children. As per the WHO report on 10 January 2023, there are a total of 176 vaccines in clinical trials and 199 in pre-clinical development for adults, with very limited availability for children (https://www.who.int/publications/m/item/draft-landscape-of-covid-19-candidate-vaccines, accessed on 30 November 2022). As per the WHO dashboard on 31 January 2023, more than 13 billion vaccine doses have been administered (https://covid19.who.int/, accessed on 30 November 2022). To fill the gap in vaccine coverage and prevent community transmission, the FDA approved the Pfizer vaccine for adolescents aged 12–17 years and children in the age group of 5–11 years under EUA. As per CDC guidelines, children above 5 years of age are eligible for full vaccination. Clinical trials are still ongoing to fully approve Pfizer and other alternative vaccines. As per CDC data, in the USA, 1.55 million children of more than 5 years of age have received at least one dose of vaccine since 18 June 2022. Recent COVID-19 vaccination status in USA according to CDC is shown in Table 1. The percentages of the populations in the USA and other countries that has received the primary series of vaccination, including children, are listed in Table 2. The CDC recommends that everyone 6 months and older get a COVID-19 vaccine, and booster for 5 years and older, as shown in Table 3. In adolescents, the Pfizer vaccine dose is equivalent to adults as 30 mg and given in two doses 21 days apart. However, in children, a 10 mg dose is administered to avoid the adverse side effects of using a higher dose. A clinical trial using Pfizer’s 10 mg dose vaccine in children aged 5–11 years has shown it to be safe, immunogenic, and with equivalent neutralization titers as seen in adolescents with an efficacy around 90% [80]. Both mRNA vaccines’ description is shown in Table 3. Three doses of the Moderna vaccine for the primary series of vaccination are recommended for immunocompromised individuals. If administered in two doses, it is given four weeks apart (28 days), and in the case of three doses, there should be a gap of a minimum of one month. The Pfizer-BioNTech vaccine was first approved in December 2020 and later fully licensed on 23 August 2021, while the Moderna vaccine was first approved in December 2020 and fully licensed in January 2022. These vaccines are administered via an intramuscular injection into the deltoid muscle. Recently, Pfizer BioNTech’s and Moderna’s bivalent vaccines also received approval under EUA for use in children above 5 years of age to receive their single dose as a booster. These bivalent boosters consist of Wuhan spike mRNA and BA.5 spike mRNA in a 1:1 ratio. These bivalent vaccines provide higher or equivalent neutralizing antibody titers against the Omicron subvariant [81,82,83]. So far, no bivalent vaccine is available for kids aged 6 months to 4 years. A summary of the currently available bivalent vaccines is provided in Table 4. Children responded in a similar manner or better post-vaccination than adults. Both antibody binding and neutralizing antibody responses to SARS-CoV-2 in both very young children and adolescents exceeded those in adults [84,85]. Besides these two mRNA-based vaccines, Sinovac-CoronaVac and BBIBP-CorV, which are based on the inactivated virus, were approved in China for children aged 3–17 years [86,87]. Similarly, Covaxin (BB152), which is also based on an inactivated virus strategy and developed by Bharat Biotech, India, was approved in India under the EUA category [88,89]. Alternative vaccines which are under clinical trials include the Adenovirus-based vaccine Ad5-nCoV in China for 6–17 years old children, and the DNA vaccine ZyCoV-D in India for children aged 12–17 years [90]. mRNA vaccination has been reported to be minimally or moderately affected by variants of concern (VOCs), such as Alpha (B.1.17), Delta (B.1.617.2), and Beta (B.1.351) variants. However, the recent emergence of Omicron caused serious concerns about vaccine efficacy and neutralization potential. A study was conducted in fully vaccinated children between the ages of 5–11 years and 12–17 years to evaluate vaccine efficacy during the Omicron variant from December 2021 to January 2022. They observed that vaccine efficacy was reduced by 66–51% in adolescents (12–17) and 68–12% among children aged 5–11 years, which shows that children in the age group of 5–11 years were more affected by the loss of vaccine efficacy than adolescents [91]. Similarly, another report showed that neutralization by post-vaccination serum from children was reduced against VOCs such as Alpha, Beta, Gamma, and Delta, and the most reduction was seen in the Omicron variant [41].

## 5. Conclusions

The primary aim of this review is to provide the reader with a detailed picture of the current COVID-19 infections in children and highlight various issues that play a role in the decision process for the vaccination of children. Despite the high percentage of efficacy observed in the case of mRNA vaccines, and large range of protection they offer against various variants of SARS-CoV-2, the vaccination rate among children is still low. Children’s immune system is still at a naïve stage of development, and several differences exist in the immune response of children vs. adults during COVID-19 infections. The role of children in the spread of COVID-19 was initially less appreciated, as the overall infection rate in them was low, and they often remained asymptomatic or with milder symptoms. However, in the case of the Omicron variant, the number of cases among children was seen to increase. Additionally, COVID-19 associated multisystem inflammation cases have been reported in children (MIS-C), and a study conducted in the Danish population reported a reduced occurrence of MIS-C in a cohort of vaccinated children [78]. All these factors have made it imperative for governments in various countries to ensure absolute vaccine coverage for the different age groups of children in their population. This is more important, as children can act as potential carriers of the virus and contribute to the transmission to high-risk people within their families. Their infections may go unnoticed otherwise, due to mild symptoms.

The growing need to increase the rate of vaccination in children depends, to a large extent, on the approval of their parents. While misinformation about vaccination has hindered the acceptance of COVID-19 vaccines among parents, a shortage of studies on the possible long-term side effects on children’s health also contributes to the challenge. In addition, other important factors affecting vaccine coverage in children are the availability of vaccines in different geographic areas, and the effect of existing endemic infectious disease on the immune response to vaccines, e.g., helminth immunomodulation in sub-Saharan Africa. The spread of relevant and correct information by healthcare practitioners and national agencies about vaccines will help more people accept and realize the benefits of childhood vaccination and help control the transmission of COVID-19 to a large extent.

## Figures and Tables

**Figure 1 vaccines-11-00418-f001:**
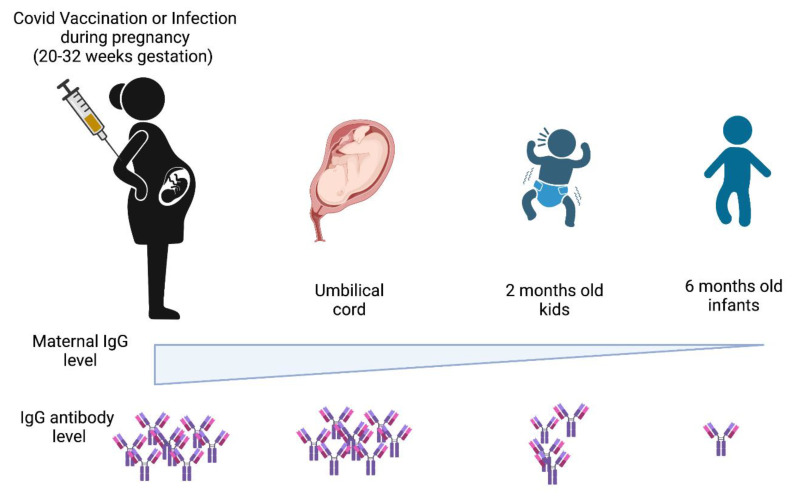
Levels of maternal IgG antibody transferred from mother to infant wanes during 0–6 months after birth. Maternal anti-SARS-CoV-2 IgG antibodies acquired either by natural infection or vaccination are transferred to the fetus via placenta. The level of maternal antibodies decreases with the infant’s age and lasts up to 6 months. SARS-CoV-2 infection or COVID-19 vaccination during 20–32 weeks of gestation generated the highest antibody titer in pregnant women. This gradually declines when tested in umbilical cord samples at the time of birth, and then further declines at 2 months after birth as observed in infant serum samples and reaches a minimal amount at the age of 6 months, as represented in Figure 1.

**Figure 2 vaccines-11-00418-f002:**
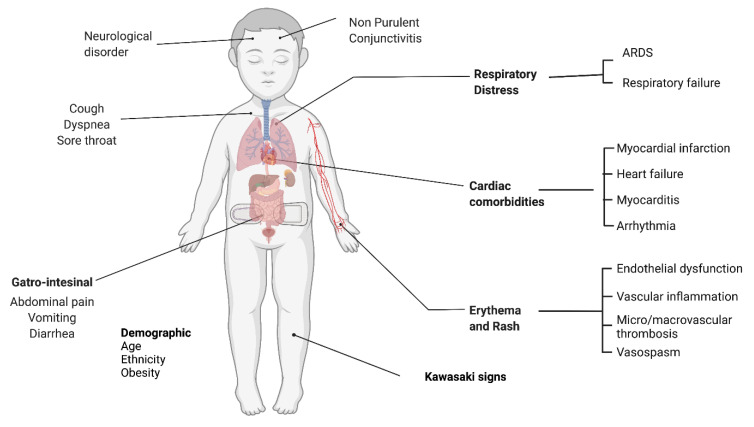
Organs affected in MIS-C immunopathological disorder in children. This disorder affects multiple organs and shows a systemic immune response. It mainly causes respiratory distress, cardiac comorbidities, gastrointestinal upset, and vascular dysfunction. This disorder is associated with demographic features such as various age form, ethnicity, and obesity level.

**Table 1 vaccines-11-00418-t001:** COVID-19 Vaccination status in the United States as per CDC records who have completed their primary vaccination series.

Fully Vaccinated People	Percent of US Population
Total	68.7%
Population ≥ 5 Years of Age	72.8%
Population ≥ 12 Years of Age	76.9%
Population ≥ 18 Years of Age	78.5%
Population ≥ 65 Years of Age	93.7%

**Table 2 vaccines-11-00418-t002:** COVID-19 Vaccinations status in the global context as per WHO records who have completed their primary vaccination series (https://covid19.who.int/table, accessed on 30 November 2022).

Fully Vaccinated People	Per 100 Persons
World	64.51
USA	68.42
India	68.94
France	78.9
Germany	76.37
Brazil	79.52
Japan	81.54
Republic of Korea	87.17
Italy	82.95
The United Kingdom	74.59
Russian Federation	53.6

**Table 3 vaccines-11-00418-t003:** COVID-19 vaccine doses and status for different age groups.

Pfizer-BioNTech Vaccine Authorized Age	Dose	Usage Status
16 years and older (Comirnaty brand name)	2 dose primary series (30 µg/dose)	Fully licensed
12–16 years old	2 dose primary series. (30 µg/dose)	Fully approved
5–11 years old	2 dose primary series. (10 µg/dose)	Under EUA and not fully approved
6 months- 4 years old	3 dose primary series (3 µg/dose)	Under EUA and not fully approved
Moderna vaccine authorized age		
17 years and older (Spikevax brand name)	2 dose primary series (100 µg/dose)	Fully licensed
12–16 years old	2 dose primary series (100 µg/dose)	Under EUA
6–11 years old	2 dose primary series (50 µg/dose)	Under EUA
Novovax vaccine authorized age		
12 years and older	2 dose primary series (0.5 mL/dose)	Under EUA
Coronavac-Sinovac and BBIBP-CorV (Sinopharm)		
3–17 years	2 doses (0.5 mL/dose)	Approved by China officials [92,93,94]
Ad5-nCoV (CanSino)		
6–17 years		Phase 2b clinical trial in China [95]
Covavax from Novavax company		
12–17 years	10 doses (0.5 mL/dose)	
6 months to 11 years	2 doses of 5 μg	https://clinicaltrials.gov/ct2/show/NCT05468736, accessed on 30 November 2022
Covaxin (BB152) by Bharat Biotech		
12–17 Years		Approved by Indian Officials
2–18		Phase 2–3 clinical trial in India [96]
Corbevax	2 doses (0.5 mL/dose)	[97]
5–17-year-old		
ZyCoV-D (Zydus Cadila)		Phase 3 clinical trial in India [98]
12–17 years old		
ChAdOx1 nCov-19 (AZD1222)		Phase 2 clinical trial in UK
6–17 years old	2 doses (5 × 10^10^) viral particle	[99]

**Table 4 vaccines-11-00418-t004:** Bivalent Vaccines approved for children so far are listed below.

Pfizer-BioNTech Bivalent Vaccine Authorized Age	Dose	Usage Status
16 years and older	1 dose (30 µg/dose)	Under EUA
12–16 years old	1 dose (30 µg/dose)	Under EUA
5–11 years old		Under EUA
6 months–4 years old		Not approved yet
Moderna vaccine bivalent authorized age		
12 years and older	1 dose (50 µg/dose)	Under EUA

## Data Availability

All authors are ready to share the information which we have provided in this article.

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
