# Peer review of "Children’s SARS-CoV-2 Infection and Their Vaccination"

_vaccines, 2023, doi:10.3390/vaccines11020418_

Round 1
Reviewer 1 Report
It has been a pleasure to revise the manuscript “Children’s Covid-19 infection and their vaccination: A need for filling the gap in vaccine coverage”, submitted for publication in Vaccine. In general, I consider that we are in the presence of a useful and interesting work. Despite of the general good quality of this review, I have some comments and suggestions for the authors. I am going to mention those comments and suggestions in the same order they appear in the manuscript:
- At the beginning of the introduction the authors state “SARS-CoV-2 has displayed a very high rate of spread since December 2019, resulting in an enormous global death toll. A total of 637 million confirmed cases and 6.6 million deaths due to SARS-CoV-2 infections have been reported until now, as per the World health organization (WHO) epidemiological report. [1].” Reference 1 was published in 2020. Therefore, it does not allow documenting the cited figures that must correspond to the contemporaneity of the manuscript. This reference also corresponds to an article in Science and not to a document issued by WHO.
- The authors analyze that a group of differences in innate and adaptive immune responses reflect why children show a distinct immune response post-SARS-CoV2 infection than adults. The analysis of these differences is correct, but other factors must be taken into account. From a more holistic perspective, a wider range of reasons, or a combination of them, have been alluded in several papers to explain the lower susceptibility of minors to SARS-CoV-2 infection and to the development of severe forms of COVID-19 and should be addressed by the authors . Among other factors present in pediatric ages, the following should be mentioned: (i) Lower intensity of exposure to SARS-CoV-2 (since the family dynamics established during the pandemic are intended to protect the youngest); (ii) Higher frequency of recurrent and concurrent viral infections (these infections can induce a state of activation of the innate immune system); (iii) Modulation of the inflammatory component of host immune responses to helminthic infections, which is more frequent among pediatric ages. Among other factors that increase with age, the following should be mentioned: (i) Alterations in endothelial functions and coagulation present in older people; (ii) Changes in the density and affinity of the enzyme converting angiotensin 2 and the transmembrane serine protease 2 enzyme in epithelial cells of the mucosa of the respiratory system; (iii) Increased immunosenescence and chronic inflammation; (iv) Higher prevalence of comorbidities.
- Authors should refer to multi-inflammatory syndrome in children with caution. In the text, between lines 194 and 195 they state “Thus, children in the omicron wave had a lesser risk of MIS-C development than in earlier delta or alpha variant waves”. This statement is correct; however, in the conclusions, between lines 275 and 279 they mention as a factor in favor of prompt immunization of children “Additionally, severe health complication of MIS-C was reported in children for the first time in 2022. MIS-C cases present with multisystem inflammation affecting various organ systems, including the heart. All these factors have made it imperative for governments in various countries to ensure absolute vaccine coverage for the different age groups of children in their population”. Certainly, the MIS-C factor should be mentioned clearly and without alarmism.
- The author makes an interesting analysis of the factors necessary to increase the rate of COVID-19 vaccination in children. However, we believe that at least two other factors should be taken into account: the availability of vaccines in all geographic areas and the need to conclude studies that allow knowing the effectiveness of those vaccines in scenarios where their immunogenicity could be affected by the high endemicity of other infectious diseases; for example, by helminthic immunomodulation in sub-Saharan Africa.
Author Response
Comments and Suggestions for Authors:
It has been a pleasure to revise the manuscript “Children’s Covid-19 infection and their vaccination: A need for filling the gap in vaccine coverage”, submitted for publication in Vaccine. In general, I consider that we are in the presence of a useful and interesting work. Despite of the general good quality of this review, I have some comments and suggestions for the authors. I am going to mention those comments and suggestions in the same order they appear in the manuscript:
- At the beginning of the introduction the authors state “SARS-CoV-2 has displayed a very high rate of spread since December 2019, resulting in an enormous global death toll. A total of 637 million confirmed cases and 6.6 million deaths due to SARS-CoV-2 infections have been reported until now, as per the World health organization (WHO) epidemiological report. [1].” Reference 1 was published in 2020. Therefore, it does not allow documenting the cited figures that must correspond to the contemporaneity of the manuscript. This reference also corresponds to an article in Science and not to a document issued by WHO.
Response: Thanks for your valuable comment and noticing this. We have corrected the reference and added a few more references to support this line. We also updated the number of confirmed cases and deaths as per the WHO website. (Link is also mentioned in the text).
- The authors analyze that a group of differences in innate and adaptive immune responses reflect why children show a distinct immune response post-SARS-CoV2 infection than adults. The analysis of these differences is correct, but other factors must be taken into account. From a more holistic perspective, a wider range of reasons, or a combination of them, have been alluded in several papers to explain the lower susceptibility of minors to SARS-CoV-2 infection and to the development of severe forms of COVID-19 and should be addressed by the authors . Among other factors present in pediatric ages, the following should be mentioned: (i) Lower intensity of exposure to SARS-CoV-2 (since the family dynamics established during the pandemic are intended to protect the youngest); (ii) Higher frequency of recurrent and concurrent viral infections (these infections can induce a state of activation of the innate immune system); (iii) Modulation of the inflammatory component of host immune responses to helminthic infections, which is more frequent among pediatric ages. Among other factors that increase with age, the following should be mentioned: (i) Alterations in endothelial functions and coagulation present in older people; (ii) Changes in the density and affinity of the enzyme converting angiotensin 2 and the transmembrane serine protease 2 enzyme in epithelial cells of the mucosa of the respiratory system; (iii) Increased immunosenescence and chronic inflammation; (iv) Higher prevalence of comorbidities.
Response: Thanks for your valuable suggestions. We have included all the factors suggested by the reviewer in our revised manuscript and added references to support these findings.
- Authors should refer to multi-inflammatory syndrome in children with caution. In the text, between lines 194 and 195 they state “Thus, children in the omicron wave had a lesser risk of MIS-C development than in earlier delta or alpha variant waves”. This statement is correct; however, in the conclusions, between lines 275 and 279 they mention as a factor in favor of prompt immunization of children “Additionally, severe health complication of MIS-C was reported in children for the first time in 2022. MIS-C cases present with multisystem inflammation affecting various organ systems, including the heart. All these factors have made it imperative for governments in various countries to ensure absolute vaccine coverage for the different age groups of children in their population”. Certainly, the MIS-C factor should be mentioned clearly and without alarmism.
Response- We have revised the entire paragraph and reframed it. The first MISC cases were reported in April 2020, so we have corrected the year. We have modified the conclusion as well.
- The author makes an interesting analysis of the factors necessary to increase the rate of COVID-19 vaccination in children. However, we believe that at least two other factors should be taken into account: the availability of vaccines in all geographic areas and the need to conclude studies that allow knowing the effectiveness of those vaccines in scenarios where their immunogenicity could be affected by the high endemicity of other infectious diseases; for example, by helminthic immunomodulation in sub-Saharan Africa.
Response: Thanks for your valuable comments. We have discussed those factors in our revised manuscript.
Reviewer 2 Report
The present work by Gupta and colleagues aims to provide a summary of main aspects of SARS-CoV-2 infection in children and the state-of-the-art of vaccination in paediatric population, in terms of immunization efficacy, safety and parents' acceptance. The study data should help in improve vaccination policies and coverage in children, to prevent severe complications and limit their role as asymptomatic source of infection.
The paper is quite well written, even it results somehow colloquial and poorly structured: the use of forms like "the need of the hour" should be avoided, as well as verbs like "illustrated" in place of "showed" are not appropriate.
Regarding the text organization, the paragraph "Vaccine hesitancy" should be moved after those on infection, to make it more congruent. Moreover, no study description is reported: how authors searched the literature? What inclusion criteria they used? Such information are fundamental to understand work contents.
Here follow more detailed observation.
Title
Line 2: Covid should be all in capital letters; in addiotin, COVID-19 is the disease: it is appropriate to use COVID-19 alone or SARS-CoV-2 infection, here and also in the text. Moreover, the sentence "A need for filling the gap in vaccine coverage" sounds littly connected to study data and outcomes.
Abstract
Line 18: are aged and immunocompromised individuals at higher risk of infection or severe outcomes?
Line 22: "to COVID-19" is redundant with line 19.
Introduction
Line 28: SARS-CoV-2 must be cited in extend form as first occurrence
Line 31-32: is not clear what variants are and why they are a continuous challenge, also because ref 2 is not this topic.
Line 33-35: data reported in refs 3 and 4 were collected ut to May 5th, 2020, being obsolete for the current situation.
Line 35-37: review in ref 5 has poor data to support infection mild course and low fatality rate; CDC, ECDC and other health agencies provide more details on their website, derived from literature.
Line 42-44: "In addition, children can form a point of contact with COVID-19 infection from their family members, which might constitute individuals in the high-risk group." It is not clear the concept of this sentence: are children source of infection for family members, especially those at higher risk, or they could transmit infection from family to community?
Vaccine hesitancy in Parents to have their children vaccinated
Line 53-59: too general sentences, with no suporting refernces.
Line 59-61: what did these studies report? It would be better to propose refernces only after data description.
Line 62: ref 12 is about children vaccination in general, not SARS-CoV-2: "milder form" refers to COVID-19, while the paper is from 2012.
Line 62-70: cited studies are limited and their data are poorly discussed by authors.
Line 70-72: why MIS-C is reported here and not only in dedicated section? Is it misplaced?
Line 74-77: is mild disease in children linked to hesitancy? It is not very clear and refs 18-19 are about infection in children (in the very early pandemic period), not properly placed along with the related sentence.
SARS-CoV-2 infection scenario in children
Line 78: What the authors mean with "infection scenario"? Scenario is not a proper term in this context.
Line 79-91: too generic paragraph, poor references; in addition, the case study reported on Chinese families is very limited and lot of data are available on transmission, making this report unuseful.
Line 91-99: COVID-19 in pregnancy and viral vertical transmission are largely discussed in more recent papers, that should be selected for a better presentation of the topics in the review; in addition, the paragraph is too generic.
Line 97: ref 26 is insufficient to support this sentence, since the study included a small population.
Line 115: what does it mean that "younger kids are transmissible"? The virus is transmitted, kids are infected.
Line 113-122: this part is doubled.
Line 123-131: discussion and relative data are limited, long COVID is not defined and the concepts development is not linear. Sentences on the same argument must be linked.
Line 132-154: the children immune response is for sure fundamental to explain the different disease severity in paediatric population; for this reason, more details are required to let readers clearly understand mechanisms: in the present form, data are insufficient and rough.
Multi-inflammatory syndrome in children
The section is quite vague and unstructured. MIS-C description, differences with other syndromes, association with COVID-19, treatments and all other aspects are not clear and almost confunding. Authors at the beginning refer a link between COVID-19 and MIS-C, but in line 180-198 it seems that syndromes are not related, leading to misunderstanding. In addition, in line 184-185 B cells mutations development is reported as a difference between COVID-19 and MIS-C: it is strange, since mutations development in humans takes long time, so the mutational pattern should be a pre-existing one.
Current status of vaccination in children
A large part of the section deals with vaccine formulations and schedules, as reported in table 2, resulting redundant. In addiotion, only CDC data were reported, without any refernce to their source.
Line 256-259: this sentence as poor scientific sound
Conclusion
Conclusions are in line with the work, but there is poor scientific contribution.
References
- References 1 and 2 are incorrect
- Reference 13 is a repetition of 12
- References 56 and 64 are pre-print, for which published papers are now available
Figures
Figure 1. Is "Maternal IgG antibody..." the figure title? It should be reported in the caption. What is the difference between "Maternal IgG level" and "IgG antibody level"? Does umbilical cord refer to IgG levels at birth? Line 156: "vertical transmission" is improper, being used for infections. Caption is not very explanatory. It is not necessary to report software here.
Figure 2. Is "Multi-system inflammatory..." the figure title? Are symptoms and syndromes reported MIS-C causes, risk factors or consequences? Line 200: what does it mean "Organ's effect"? Caption is not very explanatory. It is not necessary to report software here.
Tables
Tables are unsufficient, reporting only CDC data and they are also not very clear.
Author Response
Responses to reviewer 2:
No study description is reported: how authors searched the literature? What inclusion criteria they used? Such information is fundamental to understand work contents
Response: We have added a literature Search section in our revised manuscript. All relevant studies described in this review are from peer-reviewed scientific publications. Most current studies delineate adult COVID-19 infection and/or vaccine responses with limited data on how children respond post-infection and vaccination and on newly emerging circulating VOCs. To find available literature on vaccine-induced pediatric research, we did PubMed, Official websites such as FDA, CDC, and WHO, and clinical trials. The search was based on a query using these keywords “(B cell) OR (Humoral Immune response) AND (Neutralization) AND (Antibody binding) AND (Infection) AND (Vaccine) AND (MISC) AND (Bivalent Vaccine).” We did not keep a filter for our search by either language or type of publication.
Title
Line 2: Covid should be all in capital letters; in addition, COVID-19is the disease: it is appropriate to use COVID-19 alone or SARS-CoV-2 infection, here and also in the text. Moreover, the sentence “A need for filling the gap in vaccine coverage" sounds littly connected to study data and outcomes:
Response: Thanks for your valuable suggestions. We have modified the title to “Children’s SARS-CoV2 infection and their vaccination”.
Abstract
Line 18: are aged and immunocompromised individuals at higher risk of infection or severe outcomes?
Response: Thanks for your valuable question. The aged and immunocompromised individuals are at high risk and show severe disease outcomes too. This sentence has been re-framed to reflect this.
Line 22: "to COVID-19" is redundant with line 19.
Response: We have noted the redundancy in the two sentences and line 19 has been changed to remove the redundancy.
Introduction
Line 28: SARS-CoV-2 must be cited in extend form as first occurrence
Response: Thanks for your valuable suggestion. The suggestion has been incorporated in our revised manuscript.
Line 31-32: is not clear what variants are and why they are a continuous challenge, also because ref 2 is not this topic.
Response: Thanks for your valuable suggestion. We have now mentioned the different variants of SARS-CoV-2, which is a challenge for the healthcare system, and corrected the associated references.
Line 33-35: data reported in refs 3 and 4 were collected ut to May 5th, 2020, being obsolete for the current situation.
Response: Thanks for your valuable suggestion. We have included recent references in our revised manuscript.
Line 35-37: review in ref 5 has poor data to support infection mild course and low fatality rate; CDC, ECDC and other health agencies provide more details on their website, derived from literature.
Response: Thanks for your valuable suggestion. We have revised the references supporting these findings in our revised manuscript.
Line 42-44: "In addition, children can form a point of contact with COVID-19 infection from their family members, which might constitute individuals in the high-risk group." It is not clear the concept of this sentence: are children source of infection for family members, especially those at higher risk, or they could transmit infection from family to community?
Response: Thanks for your valuable comment. The sentence has been corrected to clarify its concept: children can easily spread SARS-CoV2 infection to high-risk group individuals in their family since the children might not show any severe symptoms and, therefore, parents and family members might not notice their infections.
Vaccine hesitancy in Parents to have their children vaccinated
Line 53-59: too general sentences, with no supporting references.
Response: The supporting references for these sentences have now been included in the revised manuscript.
Line 59-61: what did these studies report? It would be better to propose references only after data description.
Response: Thanks for your valuable suggestion. The observations of the three studies have now been discussed in the section.
Line 62: ref 12 is about children vaccination in general, not SARS-CoV-2: "milder form" refers to COVID-19, while the paper is from 2012.
Response: Thanks for your valuable comment. The section has been revised and line 61-62 and reference 12 has been removed.
Line 62-70: cited studies are limited, and their data are poorly discussed by authors.
Response: Thanks for your valuable comment. The concepts that were discussed in line 62-66 were somewhat repetitive with the description of studies of references 61 to 64 (updated numbers) and therefore the section has been re-structured. References supporting lines 66-68 have also been included.
Line 70-72: why MIS-C is reported here and not only in dedicated section? Is it misplaced?
Response: The line has now been shifted to the appropriate section in MIS-C
Line 74-77: is mild disease in children linked to hesitancy? It is not very clear and refs 18-19 are about infection in children (in the very early pandemic period), not properly placed along with the related sentence.
Response: The intention of this line was to highlight that children act as asymptomatic carriers of the SARS-Co-V2 virus. However, this information was found to be repetitive and therefore it has been removed from the section.
SARS-CoV-2 infection scenario in children
Line 78: What the authors mean with "infection scenario"? Scenario is not a proper term in this context.
Response: We have remove the word scenario and revise the heading as “ SARS-CoV-2 infection in children”
Line 79-91: too generic paragraph, poor references; in addition, the case study reported on Chinese families is very limited and lot of data are available on transmission, making this report unuseful.
Response: We have added recent references to support this finding.
Line 91-99: COVID-19 in pregnancy and viral vertical transmission are largely discussed in more recent papers, that should be selected for a better presentation of the topics in the review; in addition, the paragraph is too generic.
Response: We have added recent references to support this finding.
Line 97: ref 26 is insufficient to support this sentence, since the study included a small population.
Response: We have added recent references to support this finding.
Line 115: what does it mean that "younger kids are transmissible"? The virus is transmitted, kids are infected.
Response: We have reframed the sentence. Disease are transmissible by kids.
Line 113-122: this part is doubled.
Response: We have removed the redundant part.
Line 123-131: discussion and relative data are limited, long COVID is not defined and the concepts development is not linear. Sentences on the same argument must be linked.
Response: We have removed long COVID part as it was beyond the scope of this review.
Line 132-154: the children immune response is for sure fundamental to explain the different disease severity in paediatric population; for this reason, more details are required to let readers clearly understand mechanisms: in the present form, data are insufficient and rough.
Response: We have added recent references to support this finding.
Multi-inflammatory syndrome in children
The section is quite vague and unstructured. MIS-C description, differences with other syndromes, association with COVID-19, treatments and all other aspects are not clear and almost confunding. Authors at the beginning refer a link between COVID-19 and MIS-C, but in line 180-198 it seems that syndromes are not related, leading to misunderstanding. In addition, in line 184-185 B cells mutations development is reported as a difference between COVID-19 and MIS-C: it is strange, since mutations development in humans takes long time, so the mutational pattern should be a pre-existing one.
Response: We have revised the paragraph and removed the confusing part to make it clear
Current status of vaccination in children
A large part of the section deals with vaccine formulations and schedules, as reported in table 2, resulting redundant. In addition, only CDC data were reported, without any reference to their source.
Response: We have revised the tables and added other vaccines available globally.
Line 256-259: this sentence as poor scientific sound
Response: We have revised this sentence in our revised manuscript.
Conclusion
Conclusions are in line with the work, but there is poor scientific contribution.
Response: We have revised and elaborated the conclusion to make it more scientifically relevant.
References
- References 1 and 2 are incorrect
Response: We have corrected the references now.
- Reference 13 is a repetition of 12
Response: Correction done as per suggestion.
- References 56 and 64 are pre-print, for which published papers are now available
Response: Thanks for the information. We have cited the original paper now.
Figures
Figure 1. Is "Maternal IgG antibody..." the figure title? It should be reported in the caption. What is the difference between "Maternal IgG level" and "IgG antibody level"? Does umbilical cord refer to IgG levels at birth? Line 156: "vertical transmission" is improper, being used for infections. Caption is not very explanatory. It is not necessary to report software here.
Response: We have removed the figure title and added that caption. Also, remove the software part. We also elaborate on figure legend. Remove vertical transmission words. Yes, the umbilical cord sample is analyzed at the time of birth.
Figure 2. Is "Multi-system inflammatory..." the figure title? Are symptoms and syndromes reported MIS-C causes, risk factors or consequences? Line 200: what does it mean "Organ's effect"? Caption is not very explanatory. It is not necessary to report software here.
Response: We have removed the figure title and added that caption. Also, remove the software part. We also elaborated on the figure legend.
Tables
Tables are unsufficient, reporting only CDC data and they are also not very clear.
Response: We have revised the tables and added other vaccines available globally. We also added references to support that.
Reviewer 3 Report
This review provides a detailed picture of current COVID-19 infections in children and highlights various issues that play a role in the decision process for childhood vaccination. It can be accepted for the publication after some major revisions.
1. The motivation and novelty of the article should be more explained in the introduction section.
2. Compare your results with others existing in the literature.
3. Give more informations about the role of adaptive immunity in Children's SARS-CoV-2 infection.
4. Check and unify the citation of the references.
5. There are some typos. The authors should carefully read the manuscript.
Author Response
1.The motivation and novelty of the article should be more explained in the introduction section.
Response: We have now included more information which helps to better explain the motivation and novelty for the current work in the introduction section which has been shown in highlighted region.
- Compare your results with others existing in the literature.
Response: Our study has focused on the several factors which include COVID-19 infections in children, their immune responses, complications arising in children and the available vaccine candidates. While many studies have focused on one or more factors that we have also focused, we present here a more comprehensive summary.
- Give more information about the role of adaptive immunity in Children's SARS-CoV-2 infection.
Response: We have included more information on the role of adaptive immunity in children during the SARS-CoV-2 infection under the section of ‘SARS-CoV-2 infection in children’ which can be seen in highlighted section.
- Check and unify the citation of the references.
Response: We have re-checked the citations for references used in the manuscript and made appropriate corrections.
- There are some typos. The authors should carefully read the manuscript.
Response: We have corrected the typographical errors in the manuscript.
Round 2
Reviewer 1 Report
I endorse.
Author Response
Response: Thanks for the evaluation and for endorsing our manuscript for possible publication.
Reviewer 2 Report
Thank you for your work in modifing the paper.
Still, more improvement, mainly in the text form, are necessary to make it more readable and understandable.
Author Response
Response: Thanks for your nice comment. We have corrected the grammatical and typographical errors we found throughout the manuscript and modified the sentence formation when necessary to facilitate ease in readability and understanding.
Reviewer 3 Report
The paper can be accepted.